# Development of a Nomogram Predicting the Risk of Persistence/Recurrence of Cervical Dysplasia

**DOI:** 10.3390/vaccines10040579

**Published:** 2022-04-09

**Authors:** Giorgio Bogani, Luca Lalli, Francesco Sopracordevole, Andrea Ciavattini, Alessandro Ghelardi, Tommaso Simoncini, Francesco Plotti, Jvan Casarin, Maurizio Serati, Ciro Pinelli, Alice Bergamini, Barbara Gardella, Andrea Dell’Acqua, Ermelinda Monti, Paolo Vercellini, Innocenza Palaia, Giorgia Perniola, Margherita Fischetti, Giusi Santangelo, Alice Fracassi, Giovanni D’Ippolito, Lorenzo Aguzzoli, Vincenzo Dario Mandato, Luca Giannella, Cono Scaffa, Francesca Falcone, Chiara Borghi, Mario Malzoni, Andrea Giannini, Maria Giovanna Salerno, Viola Liberale, Biagio Contino, Cristina Donfrancesco, Michele Desiato, Anna Myriam Perrone, Giulia Dondi, Pierandrea De Iaco, Simone Ferrero, Giuseppe Sarpietro, Maria G. Matarazzo, Antonio Cianci, Stefano Cianci, Sara Bosio, Simona Ruisi, Lavinia Mosca, Raffaele Tinelli, Rosa De Vincenzo, Gian Franco Zannoni, Gabriella Ferrandina, Marco Petrillo, Giampiero Capobianco, Salvatore Dessiole, Annunziata Carlea, Fulvio Zullo, Barbara Muschiato, Stefano Palomba, Stefano Greggi, Arsenio Spinillo, Fabio Ghezzi, Nicola Colacurci, Roberto Angioli, Pierluigi Benedetti Panici, Ludovico Muzii, Giovanni Scambia, Francesco Raspagliesi, Violante Di Donato

**Affiliations:** 1Department of Gynecological, Obstetrical and Urological Sciences, “Sapienza” University of Rome, 00161 Rome, Italy; innocenza.palaia@uniroma1.it (I.P.); giorgia.perniola@uniroma1.it (G.P.); margherita.fischetti@uniroma1.it (M.F.); giusi.santangelo@uniroma1.it (G.S.); alice.fracassi@uniroma1.it (A.F.); andrea.giannini@uniroma1.it (A.G.); pierluigi.benedettipanici@uniroma1.it (P.B.P.); ludovico.muzii@uniroma1.it (L.M.); violante.didonato@uniroma1.it (V.D.D.); 2Gynecological Oncology Unit, Fondazione IRCCS Istituto Nazionale dei Tumori di Milano, 20133 Milan, Italy; luca.lalli@istitutotumori.mi.it (L.L.); raspagliesi@istitutotumori.mi.it (F.R.); 3Gynecological Oncology Unit, Centro di Riferimento Oncologico–National Cancer Institute, Via F. Gallini 2, 33081 Aviano, Italy; fsopracordevole@cro.it; 4Woman’s Health Sciences Department, Gynecologic Section, Polytechnic University of Marche, 60126 Ancona, Italy; ciavattini.a@libero.it (A.C.); luca.giannella@ospedaliriuniti.marche.it (L.G.); 5Azienda Usl Toscana Nord-Ovest, UOC Ostetricia e Ginecologia, Ospedale Apuane, 54100 Massa, Italy; alessandro.ghelardi@uslnordovest.toscana.it; 6Department of Clinical and Experimental Medicine, University of Pisa, 56126 Pisa, Italy; tommaso.simoncini@med.unipi.it; 7Department of Obstetrics and Gynecology, Campus Bio-Medico University of Rome, 00128 Rome, Italy; f.plotti@unicampus.it (F.P.); r.angioli@unicampus.it (R.A.); 8Department of Obstetrics and Gynecology, ‘Filippo Del Ponte’ Hospital, University of Insubria, 21100 Varese, Italy; jvancasarin@gmail.com (J.C.); mauserati@hotmail.com (M.S.); fabio.ghezzi@uninsubria.it (F.G.); 9Ospedale di circolo Fondazione Macchi, 21100 Varese, Italy; ciropinelli88@gmail.com; 10Department of Obstetrics and Gynecology, IRCCS Ospedale San Raffaele, 20100 Milano, Italy; bergamini.alice@hsr.it; 11IRCCS S. Matteo Foundation, Department of Clinical, Surgical, Diagnostic and Paediatric Sciences, University of Pavia, 27100 Pavia, Italy; barbara.gardella@unipv.it (B.G.); arsenio.spinillo@unipv.it (A.S.); 12Gynaecology Unit, Fondazione IRCCS Ca’ Granda Ospedale Maggiore Policlinico, 20122 Milan, Italy; andrea.dellacqua1@gmail.com (A.D.); ermelinda.monti@policlinico.mi.it (E.M.); paolo.vercellini@policlinico.mi.it (P.V.); 13Division of Obstetrics and Gynecology, Cesare Magati Hospital, Azienda Unità Sanitaria Locale—IRCCS di Reggio Emilia, 42019 Scandiano, Italy; giovanni_dippolito@yahoo.it (G.D.); aguzzoli.lorenzo@ausl.re.it (L.A.); vincenzodario.mandato@ausl.re.it (V.D.M.); 14Gynecology Oncology Unit, Istituto Nazionale Tumori IRCCS “Fondazione G. Pascale”, 80131 Naples, Italy; c.scaffa@istitutotumori.na.it (C.S.); francesca.falcone3@libero.it (F.F.); s.greggi@istitutotumori.na.it (S.G.); 15Department of Obstetrics and Gynecology, S. Anna University Hospital, 44121 Ferrara, Italy; brgchr1@unife.it; 16Endoscopica Malzoni, Center for Advanced Endoscopic Gynecological Surgery, 83100 Avellino, Italy; malzonimario@gmail.com; 17Department of Woman’s and Child’s Health, Obstetrics and Gynecological Unit, San Camillo-Forlanini Hospital, 00152 Rome, Italy; salerno.giovannamaria@gmail.com; 18Department of Obstetrics and Gynecology, Ospedale Maria Vittoria, 10144 Torino, Italy; viola.liberale@istitutotumori.mi.it (V.L.); biagio.contino@aslcittaditorino.it (B.C.); 19Department of Obstetrics and Gynecology, Azienda ASL Frosinone, Ospedale S Trinità di Sora, 03039 Sora, Italy; cristina.donfrancesco@gmail.com (C.D.); micheledesiato@libero.it (M.D.); 20Gynecologic Oncology Unit, Sant’Orsola-Malpighi Hospital, 40138 Bologna, Italy; myriam.perrone@aosp.bo.it (A.M.P.); giulia.dondi@gmail.com (G.D.); pierandrea.deiaco@unibo.it (P.D.I.); 21Academic Unit of Obstetrics and Gynaecology, IRCCS Ospedale Policlinico San Martino, 16132 Genova, Italy; simone.ferrero@unige.it; 22Department of Neurosciences, Rehabilitation, Ophthalmology, Genetics, Maternal and Child Health (DiNOGMI), University of Genova, 16132 Genova, Italy; 23Department of General Surgery and Medical Surgical Specialties, Gynecological Clinic University of Catania, Via S. Sofia 78, 95124 Catania, Italy; sarpietrogiuseppe@gmail.com (G.S.); mariagraziamatarazzo@gmail.com (M.G.M.); acianci@unict.it (A.C.); 24Department of Gynecologic Oncology, Università degli studi di Messina, Policlinico G. Martino, 98122 Messina, Italy; stefanoc85@hotmail.it; 25San Paolo Hospital, Università degli Studi di Milano, 20142 Milan, Italy; sarabos2004@hotmail.it (S.B.); simona.ruisi13@gmail.com (S.R.); 26Department of Woman, Child and General and Specialized Surgery, University of Campania “Luigi Vanvitelli”, 80138 Naples, Italy; laviniamosca@gmail.com (L.M.); nicola.colacurci@unicampania.it (N.C.); 27Department of Obstetrics and Gynecology, “Valle d’Itria” Hospital, Martina Franca, via San Francesco da Paola, 74015 Taranto, Italy; raffaeletinelli@gmail.com; 28UOC Ginecologia Oncologica, Dipartimento per la Salute Della Donna e del Bambino e Della Salute Pubblica, Fondazione Policlinico Universitario A. Gemelli, IRCCS, 00168 Roma, Italy; rosa.devincenzo@unicatt.it (R.D.V.); gianfranco.zannoni@policlinicogemelli.it (G.F.Z.); gabriella.ferrandina@gmail.com (G.F.); giovanni.scambia@policlinicogemelli.it (G.S.); 29Gynecologic and Obstetric Unit, Department of Medical, Surgical and Experimental Sciences, University of Sassari, 07100 Sassari, Italy; marco.petrillo@gmail.com (M.P.); capobia@uniss.it (G.C.); dessole@uniss.it (S.D.); 30Department of Neuroscience, Reproductive Science and Dentistry, School of Medicine, University of Naples Federico II, 80131 Naples, Italy; nunziacarlea@gmail.com (A.C.); fulvio.zullo@unina.it (F.Z.); 31Studio Medico Muschiato, Loano & Finale Ligure, 17025 Loano, Italy; info@studiomedicomuschiato.it; 32Unit of Obstetrics and Gynecology, GOM of Reggio Calabria & University ‘Magna Graecia’ of Catanzaro, 88100 Catanzaro, Italy; stefanopalomba@tin.it

**Keywords:** HPV, conization, cervical dysplasia, LEEP, recurrence

## Abstract

**Background:** Cervical dysplasia persistence/recurrence has a great impact on women’s health and quality of life. In this study, we investigated whether a prognostic nomogram may improve risk assessment after primary conization. **Methods:** This is a retrospective multi-institutional study based on charts of consecutive patients undergoing conization between 1 January 2010 and 31 December 2014. A nomogram assessing the importance of different variables was built. A cohort of patients treated between 1 January 2015 and 30 June 2016 was used to validate the nomogram. **Results:** A total of 2966 patients undergoing primary conization were analyzed. The median (range) patient age was 40 (18–89) years. At 5-year of follow-up, 6% of patients (175/2966) had developed a persistent/recurrent cervical dysplasia. Median (range) recurrence-free survival was 18 (5–52) months. Diagnosis of CIN3, presence of HR-HPV types, positive endocervical margins, HPV persistence, and the omission of HPV vaccination after conization increased significantly and independently of the risk of developing cervical dysplasia persistence/recurrence. A nomogram weighting the impact of all variables was built with a C-Index of 0.809. A dataset of 549 patients was used to validate the nomogram, with a C-index of 0.809. **Conclusions:** The present nomogram represents a useful tool for counseling women about their risk of persistence/recurrence after primary conization. HPV vaccination after conization is associated with a reduced risk of CIN2+.

## 1. Introduction

During the last decade, the widespread adoption of primary and secondary preventions has dramatically reduced the incidence of cervical cancer in developed countries [1]. However, cervical cancer still represents a major health concern due to its relatively high incidence (especially in developing countries) and the severity of the disease in the advanced setting [2]. In most cases, cervical cancer is caused by a persistent infection from human papillomavirus (HPV) and develops through a series of precursor dysplastic lesions of the cervical epithelium (i.e., cervical dysplasia) [1,2,3]. The prognosis for women with cervical dysplasia who receive appropriate follow-up and treatment is excellent. Misdiagnosis or delayed diagnoses might correlate with a high risk of developing cervical cancer [1]. The mainstay of treatment for women with cervical dysplasia is conization [4]. Conization allows for the removal of the lesions located on the uterine cervix. Accumulating data highlight that conization is a simple, safe, and effective procedure in managing cervical dysplasia. However, the management of residual/recurrent dysplasia after primary treatment is often challenging [5,6]. Moreover, it is important to classify patients based on their risk of having persistent/recurrent dysplasia after primary treatment [7]. Assessing these classes of risks is useful in tailoring appropriate surveillance and determining the need for adjunctive treatments. Our study group estimated the risk of developing persistent/recurrent dysplasia in several investigations, observing that positive surgical margins, surgical techniques, high-risk HPV infection at the time of diagnosis, and HPV persistence are the main prognostic factors [6,8,9]. Our data corroborated a considerable body of literature investigating this issue [10,11,12]. However, it is difficult to estimate the risk of developing persistent/recurrent dysplasia for each patient. Here, we aim to evaluate the importance of various prognostic factors in influencing the risk of persistent/recurrent lesions. Hence, we sought to develop a prediction nomogram estimating the risk of developing persistent/recurrent cervical dysplasia after primary conization.

## 2. Materials and Methods

This is an analysis of a retrospective multi-institutional dataset of patients with cervical dysplasia treated in Italy. Institutional review board (IRB) approval was obtained. For the present study, we collected medical records of patients with newly diagnosed high-grade cervical dysplasia (HSIL/CIN2/CIN3) treated in Italy between 1 January 2010 and 31 December 2014. The main outcome measure of this research was to build a nomogram that might weigh the impact of various prognostic factors in assessing the risk of developing persistent/recurrent dysplasia. Additionally, data of patients treated between 1 January 2015 and 30 June 2016 were used to validate the model. The inclusion criteria were: (i) newly diagnosed moderate/severe cervical dysplasia, (ii) squamous cell lesions, (iii) execution of surgical excisional procedure (i.e., conization); (iv) patients with available 5-year follow-up data (for non-recurring patients; patients who recurred were included even if they did not complete the five-year follow-up period). Exclusion criteria were: (i) age < 18 years, (ii) consent withdrawal, (iii) execution of ablative procedure (because type of lesion was not confirmed), (iv) diagnosis of invasive cancer at the time of conization (because the present study focused on patients with cervical dysplasia), (v) execution of cold-knife conization (because we aimed to reduce possible confounding factors related to the execution of wide cervical excisions; mostly executed in case of suspected cervical cancer), (vi) glandular lesions, (vii) history of HIV and other conditions downregulating the immune system, (viii) ongoing pregnancy, and (ix) history of hysterectomy. Demographic details, data about HPV type(s) detected, and data on treatment for the occurrence of cervical dysplasia were retrospectively reviewed. HPV types were considered high-risk according to the data of the International Agency for Research on Cancer (IARC) [13]. During the study period, different surgeons perform procedures across the participant centers. However, no differences in the facilities available for patient care and the referral patterns of various services were present. Conization aimed to remove a cone-shaped section of the cervix surrounding the endocervical canal, which includes the entire transformation zone. The technique for laser conization and LEEP were standardized [6,8,9]. Details about surgical treatments are reported elsewhere [6,8,9]. The laser conization was executed, as opposed to LEEP, depending on the available facilities of the participant centers and surgeons’ preferences. Details regarding surgical techniques, follow-up schedule, and examinations were reported elsewhere [6,8,9].

Generally, patients were treated on an outpatient basis using local anesthesia. Procedures were performed under colposcopic guidance. During the study period, according to institutional protocols, patients were evaluated colposcopically in an outpatient clinic, preferably three (in case of positive margins) to six (in case of negative margins) months after conization. Briefly, patients had a follow-up appointment scheduled, including Pap smear, colposcopy, and colposcopic-guided biopsy if clinically indicated, every 6 months for the first 2 years and annually thereafter (until 5 years). Appendix A shows the follow-up schedule. In each center, a dedicated team of gynecologists performed all gynecological and colposcopic examinations [6,8,9]. Generally, HPV testing was performed at the first examination after conization in patients with documented high-risk HPV infections. Persistence of HPV infection was defined as the persistence of the same HPV types detected at the first clinical examination following conization (generally at 6 months). Generally, patients with a detected low-grade lesion (CIN1/LSIL) had observation only. Persistence/recurrence after conization was defined as the diagnosis of cervical dysplasia requiring secondary conization or hysterectomy. Patients who had not undergone secondary conization were considered free of recurrence. The persistence of cervical dysplasia was defined by the diagnosis of cervical dysplasia at the first evaluation following conization; patients with recurrent cervical dysplasia had at least one negative examination between conization and the diagnosis of cervical dysplasia. In the model, patients with persistent and recurrent disease were grouped together because of the similar clinical impact of these conditions. Diagnosis of low-grade cervical dysplasia (i.e., LSIL/CIN1) was not considered recurrent disease.

Data are summarized using basic descriptive statistics. The risk of developing cervical dysplasia recurrence was evaluated using Kaplan–Meir and Cox proportional hazard regression models. Hazard ratio (HR) and 95% confidence intervals (CI) were calculated for each comparison. Univariate and multivariate analyses were performed when appropriate. All covariates with a *p*-value less than 0.10 based on univariate analysis were included in the multivariate model. Duration of follow-up was counted from the date of first conization and the date of the last follow-up or secondary conization. A nomogram was built for estimating the risk of cervical dysplasia persistence/recurrence. Performance testing of the nomogram was assessed in terms of discrimination (Harrell’s C-index). C-index provides an estimate of the probability that the model will correctly identify patients who had cervical dysplasia persistence/recurrence. A nomogram is a graphical calculating device, a two-dimensional diagram designed to allow for the approximate graphical computation of a mathematical function. A nomogram consists of a set of *n* scales, one for each variable in an equation. Knowing the values of n-1 variables, the value of the unknown variable can be found; by fixing the values of some variables, the relationship between the unfixed variables can be studied. Statistical analyses were performed using GraphPad Prism version 6.0 (GraphPad Software, San Diego, CA, USA), IBM-Microsoft SPSS version 20.0 (SPSS Statistics, International Business Machines Corporation IBM 2013, Armonk, NY, USA), and R software (version 3.2.3, R Foundation for Statistical Computing, Vienna, Austria).

## 3. Results

### 3.1. Study Population

Overall, charts of 2966 women undergoing conization were retrieved. The median (range) patient age was 40 (18, 89) years. Overall, 32.7% (*n* = 969), 66.9% (*n* = 1984), and 0.4% (*n* = 13) of patients had conization for CIN2, CIN3, and cytological HSIL, respectively. Table 1 reports the baseline characteristics of the study population.

At 5-year of follow-up, 175 (6%) out of 2966 patients included developed a persistent/recurrent cervical dysplasia. Median (range) recurrence-free survival was 18 (5, 52) months. Figure 1 shows 5-year recurrence-free survival.

### 3.2. Development of a Nomogram

The role of various prognostic factors in influencing the risk of developing cervical dysplasia persistence/recurrence is reported in Table 2.

Diagnosis of CIN3 (instead of CIN2), presence of HR HPV types, positive endocervical margins, HPV persistence, and the omission of HPV vaccination after conization increased significantly (*p* < 0.05) and independently of the risk of developing cervical dysplasia persistence/recurrence. Figure 2 shows the risk of developing cervical dysplasia persistence/recurrence according to margin status (*p* = 0.002, log-rank-test).

The risk of developing cervical dysplasia persistence/recurrence according to vaccination status and patient age is presented in Appendix A, respectively. A nomogram weighting the impact of all those variables was built (Figure 3), with a C-index of 0.809. No statistically significant interaction was observed between the variables.

Appendix A displays details of the logistic model. Appendix A displays the impact of variables in predicting the risk of cervical dysplasia persistence/recurrence. Appendix A reports the risk of recurrence according to the proposed model.

### 3.3. Validation Cohort

The validation cohort included 549 patients treated between 1 January 2015 and 30 June 2016. As for the primary cohort, all women included had undergone conization for high-grade cervical dysplasia and achieved a 5-year follow-up. Overall, 38 (6.9%) patients experienced recurrence. Appendix A reports baseline characteristics of the validation cohort. Using data from the validation, we observed a C-index of 0.809. Accuracy of the model was 0.7304 (95%CI: 0.6912, 0.7671). Sensitivity, specificity, positive predictive value, and negative predictive value were 0.7436, 0.5526, 0.9572, and 0.1382, respectively. Appendix A show diagnostic accuracy and the ability to predict for cervical dysplasia persistence/recurrence, respectively.

## 4. Discussion

In the present paper, we evaluated factors predicting cervical dysplasia persistence/recurrence after primary conization. Analyzing a large amount of data from about 3000 patients collected in Italy from 2010 to 2014, we built a nomogram to help clinicians assess the risk of persistence/recurrence in patients with CIN2+ lesions. As expected, our nomogram highlighted the importance of margin status and HPV persistence. Similarly, HPV vaccination after conization seems to have a non-negligible role in influencing the risk of secondary conization, as supported by accumulating data on this issue.

Although single conization has little impact on fertility and the obstetrical journey of women, adjunctive cervical excision might correlate with several issues, in particular with the risk of preterm delivery [6,7,8,9,10,11,12]. Moreover, the persistence/recurrence of cervical dysplasia is a well-known risk factor for developing other HPV-related diseases and cancer (i.e., cervical cancer). Hence, there is a clinically unmet need to identify patients at high risk of persistence/recurrence.

In the present study, we evaluated prognostic factors for women with newly diagnosed moderate or severe cervical dysplasia (HSIL/CIN2/CIN3). The results of this investigation highlight that several prognostic factors impact the risk of developing cervical dysplasia persistence/recurrence. Preoperative variables (such as age, severity of the lesions, type of HPV involved), as well as postoperative variables (such as margin status and HPV persistence), influence the risk of developing persistent/recurrent disease. Additionally, HPV vaccination after conization is one important modifiable variable that might impact on the risk of developing persistent/recurrent cervical dysplasia. Recently, our study group and others evaluated factors predicting cervical dysplasia recurrence/persistence, corroborating the results of this research [9,10,11,12]. The results observed in these studies are consistent with the findings of the present investigation. In particular, all those factors are well-known important variables influencing outcomes of women with newly diagnosed cervical dysplasia [9,10,11,12]. However, no other studies have evaluated all these variables together, building a model to assess the risk of developing persistent or recurrent disease. For the first time in the literature, we developed a prognostic nomogram to improve 5-year risk assessment in patients after conization. Building a nomogram provided a reliable prognostic method that can be used to predict the risk of recurrence in this cluster of patients. Assessing the risk of cervical dysplasia persistence/recurrence is of paramount importance. The importance of our findings might be explained by three main reasons: (i) our analysis provides a clear method to improve appropriate counseling of patients undergoing conization; (ii) a better understanding of 5-year risk assessment is crucial to tailoring appropriate follow-up schedules, especially in settings with limited resources; and (iii) by applying this nomogram, general practitioners can triage high-risk patients in specialized centers, thus reducing the number of misdiagnoses and improving the quality of care [14,15].

Several points discussed in the present papers deserve to be addressed. (i) The present nomogram is based on data from patients treated between 2010 and 2014. Although the evaluation of this period was necessary to report a 5-year analysis, we can speculate that the prevalence and the aggressiveness of various HPV types changed in the last decade. (ii) During the study period, having HPV vaccination after conization was uncommon [16]. In our series, the prevalence of patients receiving HPV vaccination after conization was less than 5%. This latter group might be highly selected, thus not reflecting the real impact of vaccination in the whole cohort of women undergoing treatment for HPV-related disease [16]. However, several other studies confirm the benefit of HPV vaccination in women treated for HSIL/CIN 2-3 [17,18,19,20]. (iii) For these reasons, our results are not applicable in women who had received prophylactic vaccination before the diagnosis of cervical dysplasia. (iv) The present analysis did not consider several features that might influence patient outcomes. In particular, we were not able to correct our results on patient medical history, smoking history, and sexual activity [21,22,23]. All those points might influence the primary outcome measure and should be considered in everyday clinical practice. However, most of these features (indirectly) impacted the risk of HPV persistence, which is one of the variables that we included in our model [21,22,23,24,25]. (v) Our analysis excluded patients undergoing cold-knife conization. We decided to exclude those patients to avoid possible confounding factors related to concerns about reliable follow-up after cold-knife conization and Sturmdorf suture of the uterine cervix [10,26]. (vi) In the present paper, we did not consider cytological abnormalities detected before conization, which might influence the type of surgical procedure and postoperative outcomes [27,28]. Further analysis considering cytological findings are warranted. (vii) Similarly, the prognostic value of emerging biomarkers (e.g., p16 and Ki-67) is of paramount importance to assess the risk of persistence/recurrence in women with cervical dysplasia and deserves further prospective investigations [15]. (viii) High-risk HPV infection is a well-known risk factor for developing cervical dysplasia and cervical dysplasia recurrence (as observed in the present model). However, we have to point out that about 10–15% of high-grade cervical dysplasia cases are high-risk HPV-negative [8]. This point has several implications for the follow-up schedules of such patients. (ix) Duration of HPV persistence influences the risk of developing recurrent cervical dysplasia. In our analysis, the majority of patients were tested 6 months after conization. However, we have to highlight that duration of HPV persistence might be a more appropriate variable to assess the risk of disease persistence/recurrence. Further investigations focusing on weighting the impact of duration of HPV persistence are warranted. (x) In our series, as expected, margin status significantly impacted the risk of recurrence. Our study highlighted that the risk of cervical dysplasia persistence/recurrence is higher for women with positive endocervical margins than women with positive ectocervical margins. The role of ectocervical and endocervical margin status is not fully understood, nor is that of achieving wide clear margins (i.e., the distance between the margin and lesion) [29,30]. Accumulating evidence supports that patients with positive endocervical margins are at higher risk compared to patients with positive ectocervical margins [29,30]. However, positive ectocervical margin also significantly impacted the risk of developing persistent/recurrent disease. Moreover, we have to take into consideration that the severity of the dysplasia might correlate with the size of the cervical lesions [30,31,32,33]. In the present research, we developed a nomogram in order to weight the risk of developing persistent/recurrent cervical dysplasia using a dataset of 2966 patients. Another dataset of 549 patients was used to validate the nomogram. Data from the validation model are encouraging. C-index, accuracy, and sensitivity were “quite high”. Obviously, the relatively low proportion of patients with recurrence impacted this analysis. We would like to stress that further datasets are needed to corroborate our model.

The main weaknesses of the present study are related to the inherent biases of the retrospective study design. Additionally, our nomogram requires an external and independent validation before its utilization in clinical practice. The main strengths of the current investigation are: (i) the multi-institutional study design; (ii) the large sample size; and (iii) the homogeneous and long-term (5-year) analysis of outcomes.

## 5. Conclusions

In conclusion, once externally validated, the present nomogram will represent a useful tool for counselling women about their risk of recurrence after primary conization. Moreover, it could be helpful to triage high-risk patients into specialized centers and tailor more appropriate follow-up schedules. Further prospective and external validation of our nomogram are warranted. Although the adoption of primary and secondary prevention is reducing the risk of developing invasive HPV-related tumors, further (medical and educational) attempts are needed to reduce the burden of HPV-related disease.

## Figures and Tables

**Figure 1 vaccines-10-00579-f001:**
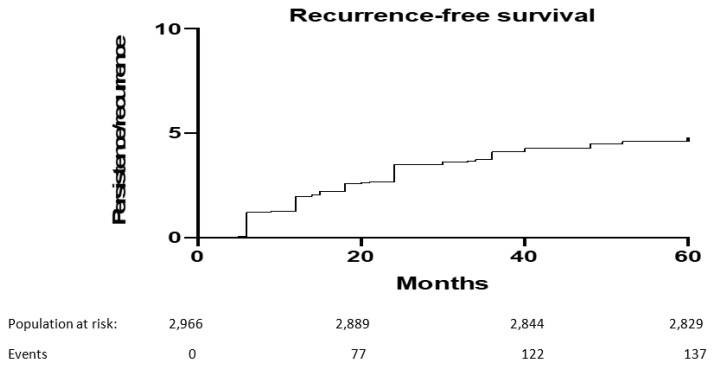
Five-year recurrence-free survival after primary conization.

**Figure 2 vaccines-10-00579-f002:**
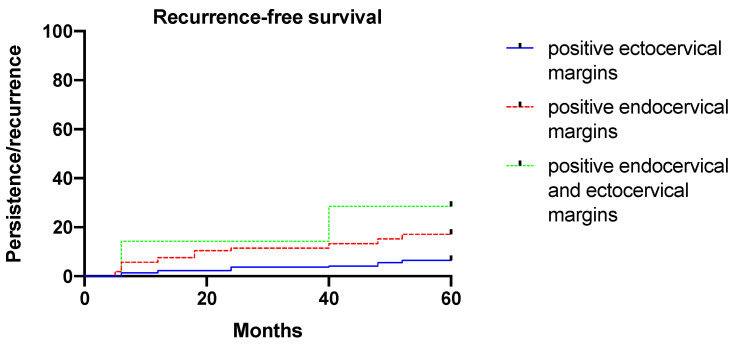
Five-year recurrence-free survival according to margin status.

**Figure 3 vaccines-10-00579-f003:**
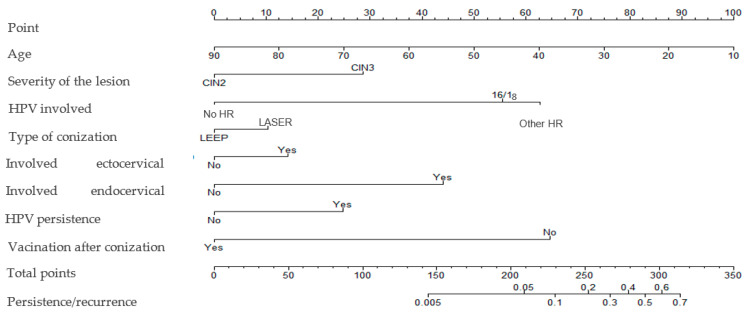
Nomogram assessing the risk of cervical dysplasia persistence/recurrence.

**Table 1 vaccines-10-00579-t001:** Baseline characteristics of the population.

Characteristics	Study Population (*n* = 2966)
Age, years	40 (18, 89)
BMI	24 (14.4, 44.0)
Menopause	
No	2373 (80.1%)
Yes	593 (19.9%)
Reason for conization	
CIN2	969 (32.7%)
CIN3	1984 (66.9%)
HSIL	13 (0.4%)
HR HPV involved *	
No	260 (14.9%)
Yes	1478 (85.1%)
Type of HPV involved	
HPV16/18	1089 (62.6%)
Other HR HPV	384 (22.1%)
No HR-HPV	260 (14.9%)
Type of conization	
Laser conization	567 (19.2%)
LEEP	2399 (80.8%)
Positive margins	
Endocervical	224 (7.5%)
Ectocervical	112 (3.8%)
Vaccination after conization	
No	2848 (96%)
Yes	118 (4%)
HPV persistence **	
No	1320 (87.1%)
Yes	196 (1.9%)

Data are reported as median (range) and number (%). Abbreviations: BMI, body mass index; CIN, cervical intraepithelial neoplasia; HSIL; high-grade squamous intraepithelial lesion; HPV, human papillomavirus; *, data on HPV involved in HSIL/CIN2+ were calculated on the basis of 1738 patients undergoing HPV testing before conization; **, data on HPV persistence were calculated on the basis of 1516 patients undergoing HPV testing after conization.

**Table 2 vaccines-10-00579-t002:** Factors impacting the risk of cervical dysplasia persistence/recurrence.

Variable	Coefficient	S.E.	Wald Z	Pr (>|Z|)
Age	−0.0453	0.0160	−2.84	0.0045 *
CIN3 vs. CIN2	1.0380	0.4157	2.50	0.0125 *
HR-HPV	−2.0121	0.8767	−2.30	0.0217 *
Other HPV types	0.2576	0.5564	0.46	0.6434
Setting: Accademic vs. Non-accademic	0.5450	0.5098	0.78	0.4515
LEEP vs. Laser conization	−0.3761	0.6244	−0.60	0.5470
Positive ectocervical margins	0.5140	0.5922	0.87	0.3854
Positive endocervical margins	1.5971	0.5678	2.81	0.0049 *
HPV persistence	0.9004	0.4047	2.22	0.0261 *
HPV vaccination after conization	−2.3436	1.0508	−2.23	0.0257 *

Abbreviations: HR, high-risk; HPV, human papillomavirus; CIN, cervical intraepithelial neoplasia. *, statistically significant.

## Data Availability

Data are available upon request.

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
