# Peer review of "Development of a Nomogram Predicting the Risk of Persistence/Recurrence of Cervical Dysplasia"

_vaccines, 2022, doi:10.3390/vaccines10040579_

Round 1

Reviewer 1 Report

The manuscript entitled “Development of a Nomogram Predicting the Risk of Persistence / Recurrence of Cervical Dysplasia” by Bogani et al. describes that the authors developed a nomogram to predict the risk of recurrence of cervical dysplasia after conization. Treatment of cervical precancer is an essential approach to eliminate cervical cancer, except for the HPV vaccination and cervical cancer screening. However, recurrence after treatment of cervical precancer can occur. The prediction of the recurrence for individual women after treatment would therefore be a meaningful and practical contribution to the field. However, the current manuscript has major shortcomings that should be addressed.

Major comments:

  1. Line 119: The authors included women with available 5-year follow-up data. Do they have any concerns about the exclusion of the severe recurrence cases who may die or loss of follow-up within the 5 years? Therefore, they may produce the selection bias in this study? How about including the women with at least one follow-up visit after treatment?
  2. It would be interesting to see the proportion of women who were persistent versus recurrence. Do they have different risk factors? Can the current model have the same performance to predict the probability of persistence versus recurrence?
  3. Missing data: Not all women have all the information on variables listed in Table 1 (for example, HPV test was available for 54% (1597/2966) women). How did the authors handle the data with missing values?
  4. Variables with interaction: In Figure 3, HPV involved has the interaction with HPV persistence, how did the authors handle this issue in their model?
  5. Discussion: The discussion needs to be drastically improved to explain the findings, rather than listing all limitations. Try to clear interpretation of the results and put them in the public health context. 

Minor comments:

Abstract:

  1. Line 69: “A cohort of patients treated between 01/012015 and 12/31/2016 was used to validate the nomogram”, but in the methods and results, the authors mentioned the deadline was 06/30/2016? Please keep them consistent.

Materials and Methods:

  1. Lines 120-121: Please clarify why they excluded the women who were executed for the ablative procedure; cervical cancer; and cold knife conization.
  2. Lines 136-145: It’s unclear about the follow-up design for the women. Could they make a schematic diagram to show when they were followed up and what kinds of tests or clinical examinations were done at each follow-up visit?

Table 1:

  1. In Table 1, please also show the variables listed in Table 2 and Figure 3, for example, what’s the distribution of women by surgery (LEEP versus laser conization), and by HR-HPV status (no HR-HPV, HPV16/18, other non-16/18 HR-HPV)? Please keep the name of the variable consistent, for example, using “HPV vaccination after conization” throughout the paper.
  2. As the authors stated in Lines 126-127, “different surgeons perform across the centre”, please show the information on the distribution of institutes and analyse the effect on the recurrence.
  3. The period of HPV persistence (i.e. for 6 months, 12 months…) might have an effect on the recurrence/persistence. Can the authors analyse them? Please also clarify the definition of HPV persistence. Is it type-specific or grouped HPV persistence?

Figures 1 and 2:

  1. It would be more straightforward to show the curves from 0, namely to show the cumulative probability of recurrence/persistence.
  2. Please add the number of women at risk as well as the number of events at certain time points.

Author Response

Reviewer #1:

The manuscript entitled “Development of a Nomogram Predicting the Risk of Persistence / Recurrence of Cervical Dysplasia” by Bogani et al. describes that the authors developed a nomogram to predict the risk of recurrence of cervical dysplasia after conization. Treatment of cervical precancers is an essential approach to eliminating cervical cancer, except for the HPV vaccination and cervical cancer screening. However, recurrence after treatment of cervical precancer can occur. The prediction of the recurrence for individual women after treatment would therefore be a meaningful and practical contribution to the field. However, the current manuscript has major shortcomings that should be addressed.

 Major comments:

 Comment 1. Line 119: The authors included women with available 5-year follow-up data. Do they have any concerns about the exclusion of the severe recurrence cases who may die or loss of follow-up within the 5 years? Therefore, they may produce a selection bias in this study? How about including the women with at least one follow-up visit after treatment?

Answer: We thank the reviewer for his/her comment. Evaluating data of women who achieve 5 years of follow-up is one of the main strengths of the present research. We excluded only women who were lost at follow-up. We clarified this point in the method section: “patients with available 5-year follow-up data (for non-recurring patients; while patients who recurred were included even if they did not complete the five-year follow-up period).”

Comment 2. It would be interesting to see the proportion of women who were persistent versus recurrence. Do they have different risk factors? Can the current model have the same performance to predict the probability of persistence versus recurrence?

Answer: To comply with the reviewer’s comment we add data about the prevalence of women with recurrent and persistent diseases. Additionally, we provide a more detailed description of the definition of disease persistence and recurrence. They are grouped since they have the same clinical impact.  

Comment 3. Missing data: Not all women have all the information on variables listed in Table 1 (for example, the HPV test was available for 54% (1597/2966) women). How did the authors handle the data with missing values?

Answer: To comply with the reviewer’s comment we added details of this in the statistical method section “Knowing the values of n-1 variables, the value of the unknown variable can be found, or by fixing the values of some variables, the relationship between the unfixed ones can be studied.”

Comment 4. Variables with interaction: In Figure 3, HPV involvement has an interaction with HPV persistence, how did the authors handle this issue in their model?

Answer: To comply with the reviewer’s comment. We asked our statistician (L.L.) to test the possible interaction between the type of HPV involved and HPV persistence. “Interaction effects occur when the effect of one variable depends on the value of another variable. Our analysis showed that there is not a statistically significant interaction between these two variables: HPV types * HPV persistence Coef= 0.843 ; SE Coef = 0.560; T-Value = 2.35; p-value = 0.501”,”Testing interaction between all the variables, we observed that no statistically significant interaction exists”. We clarified this point in the results section. 

Comment 5. Discussion: The discussion needs to be drastically improved to explain the findings, rather than listing all limitations. Try to clear interpretation of the results and put them in the public health context.

Answer: To comply with the reviewer’s comment, we improved the discussion section. 

Minor comments:

Comment 6. Abstract: Line 69: “A cohort of patients treated between 01/012015 and 12/31/2016 was used to validate the nomogram”, but in the methods and results, the authors mentioned the deadline was 06/30/2016? Please keep them consistent.

Answer: We thank the reviewer for this observation. To comply with the reviewer’s comment we corrected this point. 

Materials and Methods:

Comment 7. Lines 120-121: Please clarify why they excluded the women who were executed for the ablative procedure; cervical cancer; and cold knife conization.

Answer: To comply with the reviewer’s comment we clarified this in the method’s section. 

Comment 8. Lines 136-145: It’s unclear about the follow-up design for the women. Could they make a schematic diagram to show when they were followed up and what kinds of tests or clinical examinations were done at each follow-up visit?

Answer: To comply with the reviewer’s comment we clarified this point in the method section and added a supplemental figure showing the follow-up schedules.  

Table 1:

Comment 9. In Table 1, please also show the variables listed in Table 2 and Figure 3, for example, what’s the distribution of women by surgery (LEEP versus laser conization), and by HR-HPV status (no HR-HPV, HPV16/18, other non-16/18 HR-HPV)? Please keep the name of the variable consistent, for example, using “HPV vaccination after conization” throughout the paper.

Answer: To comply with the reviewer’s comment we added the required details in Table 1 and used “HPV vaccination after conization” throughout the whole paper

Comment 10. As the authors stated in Lines 126-127, “different surgeons perform across the center”, please show the information on the distribution of institutes and analyze the effect on the recurrence.

Answer: To comply with the reviewer’s comment we tested the variable “center” in our analysis. No statistical correlation was observed between the variable “center” and the risk of cervical dysplasia persistence/recurrence. Additionally, we compared two different settings: academic vs. non-academic. Again having the setting did not influence the risk of developing the recurrent disease (we added this variable in table 2).  

Comment 11. The period of HPV persistence (i.e. for 6 months, 12 months…) might affect the recurrence/persistence. Can the authors analyze them? Please also clarify the definition of HPV persistence. Is it type-specific or grouped HPV persistence?

Answer: We strongly agree with the comment of the reviewer. It is very clever. To comply with the reviewer’s comment we clarified this point in the method section. Most of the patients included had HPV testing at 6-month. We discussed this point in the discussion section. 

Figures 1 and 2:

Comment 12. It would be more straightforward to show the curves from 0, namely to show the cumulative probability of recurrence/persistence.

Answer: To comply with the reviewer’s comment we modified the figures, accordingly

Comment 13. Please add the number of women at risk as well as the number of events at certain time points.

 Answer: To comply with the reviewer’s comment we modified the figures, accordingly. Please note that figures are also modified according to the suggestion of reviewer 2

Reviewer 2 Report

This is a retrospective multi-institutional study including women with high-grade cervical lesions (CIN2-3) treated by conization between 01/01/2010 and 12/31/2014. A total 2,966 women undergoing primary conization were analyzed. During 5-year of follow-up, 6% of the women (175/2,966) developed a persistent / recurrent cervical dysplasia. Diagnosis of CIN3, presence of HR-HPV types, positive endocervical margins, HPV persistence, and the omission of adjuvant vaccination increased significantly and independently the risk of developing cervical dysplasia persistence/recurrence. A nomogram weighting the impact of all those variables was built.

Minor revisions

Line 77-78, Conclusions, "The present nomogram would represent a useful tool for counsel women after primary conization about their risk of persistence/recurrence."

add:

"Adjuvant HPV vaccination is associated with a reduced risk of CIN2+ after treatment."

Line 188, Figure 1, Recurrence-free survival

The plot should be "1-survival", and the Y-axis not 0-100%, but 0-10% Presistens / Recurrent risk of CIN2+, see Figure 2 in Casajuana-Pérez 2022.

Line 202, Figure 2, Recurrence-free survival accoriding to margin status.

The plot should be "1-survival", Accumulated Risk of CIN2+ in women with positive and negative margins

You should also include a figure 3 "1-suvival" Accumulated Risk of CIN2+ in vaccinated and non-vaccinated women, see Figure 2 in Casajuana-Pérez 2022.

Maybe make a figure 4 "1-suvival" Accumulated Risk of CIN2+ in women 20-39 years and 40-69 years of age.

Line 218, "Sensibility, specificity, positive predictive value, and negative predictive value" => "Sensitivity, specificity, positive predictive value, and negative predictive value"

Line 257-259, "In our series, the prevalence of patients receiving “adjuvant vaccination” was less than 5%. This latter group might be highly selected, thus not reflecting the real impact of vaccination in the whole cohort of women undergoing treatment for HPV-related disease"

add

"However, several other studies confirm the benefit of HPV vaccination in woman treated for HSIL/CIN 2-3." (Tjalma 2022, Casajuana-Pérez 2022, Di Donato)

References

Tjalma WAA, van Heerden J, Van den Wyngaert T. If prophylactic HPV vaccination is considered in a woman with CIN2+, what is the value and should it be given before or after the surgical treatment? Eur J Obstet Gynecol Reprod Biol. 2022 Feb;269:98-101. doi: 10.1016/j.ejogrb.2021.11.008. Epub 2021 Nov 11. PMID: 34979365.

https://pubmed.ncbi.nlm.nih.gov/34979365/

Casajuana-Pérez A, Ramírez-Mena M, Ruipérez-Pacheco E, Gil-Prados I, García-Santos J, Bellón-Del Amo M, Hernández-Aguado JJ, de la Fuente-Valero J, Zapardiel I, Coronado-Martín PJ. Effectiveness of Prophylactic Human Papillomavirus Vaccine in the Prevention of Recurrence in Women Conized for HSIL/CIN 2-3: The VENUS Study. Vaccines (Basel). 2022 Feb 14;10(2):288. doi: 10.3390/vaccines10020288. PMID: 35214747; PMCID: PMC8879017.

https://pubmed.ncbi.nlm.nih.gov/35214747/

Di Donato V, Caruso G, Bogani G, Cavallari EN, Palaia G, Perniola G, Ralli M, Sorrenti S, Romeo U, Pernazza A, Pierangeli A, Clementi I, Mingoli A, Cassoni A, Tanzi F, Cuccu I, Recine N, Mancino P, de Vincentiis M, Valentini V, d'Ettorre G, Della Rocca C, Mastroianni CM, Antonelli G, Polimeni A, Muzii L, Palaia I. HPV Vaccination after Primary Treatment of HPV-Related Disease across Different Organ Sites: A Multidisciplinary Comprehensive Review and Meta-Analysis. Vaccines (Basel). 2022 Feb 4;10(2):239. doi: 10.3390/vaccines10020239. PMID: 35214697; PMCID: PMC8879645.

https://pubmed.ncbi.nlm.nih.gov/35214697/

Author Response

Reviewer #2:

This is a retrospective multi-institutional study including women with high-grade cervical lesions (CIN2-3) treated by conization between 01/01/2010 and 12/31/2014. A total of 2,966 women undergoing primary conization were analyzed. During 5-year of follow-up, 6% of the women (175/2,966) developed a persistent/recurrent cervical dysplasia. Diagnosis of CIN3, presence of HR-HPV types, positive endocervical margins, HPV persistence, and the omission of adjuvant vaccination increased significantly and independently the risk of developing cervical dysplasia persistence/recurrence. A nomogram weighting the impact of all those variables was built.

Minor revisions

Comment 1: Line 77-78, Conclusions, "The present nomogram would represent a useful tool for counsel women after primary conization about their risk of persistence/recurrence." add:

 "Adjuvant HPV vaccination is associated with a reduced risk of CIN2+ after treatment."

Answer: To comply with the reviewer’s comment we modified the sentence, accordingly. 

Comment 2: Line 188, Figure 1, Recurrence-free survival The plot should be "1-survival", and the Y-axis, not 0-100%, but 0-10% Persistent / Recurrent risk of CIN2+, see Figure 2 in Casajuana-Pérez 2022. Line 202, Figure 2, Recurrence-free survival according to margin status. The plot should be "1-survival", Accumulated Risk of CIN2+ in women with positive and negative margins You should also include a figure 3 "1-survival" Accumulated Risk of CIN2+ in vaccinated and non-vaccinated women, see Figure 2 in Casajuana Pérez 2022. Maybe make a figure 4 "1-survival" Accumulated Risk of CIN2+ in women 20-39 years and 40-69 years of age.

Answer: To comply with the reviewer’s comment we modified the figures (also according to the comments of reviewer 1). Additionally, in the Supplemental material, we added the required new figures. 

Comment 3: Line 218, "Sensibility, specificity, positive predictive value, and negative predictive value" => "Sensitivity, specificity, positive predictive value, and negative predictive value"

Answer: To comply with the reviewer’s comment we modified the sentence, accordingly.

Comment 4: Line 257-259, "In our series, the prevalence of patients receiving “adjuvant vaccination” was less than 5%. This latter group might be highly selected, thus not reflecting the real impact of vaccination in the whole cohort of women undergoing treatment for HPV-related disease" add "However, several other studies confirm the benefit of HPV vaccination in women treated for HSIL/CIN 2-3." (Tjalma 2022, Casajuana-Pérez 2022, Di Donato)

References

Tjalma WAA, van Heerden J, Van den Wyngaert T. If prophylactic HPV vaccination is considered in a woman with CIN2+, what is the value, and should it be given before or after the surgical treatment? Eur J Obstet Gynecol Reprod Biol. 2022 Feb;269:98-101. doi: 10.1016/j.ejogrb.2021.11.008. Epub 2021 Nov 11. PMID: 34979365. http://pubmed.ncbi.nlm.nih.gov/34979365/

Casajuana-Pérez A, Ramírez-Mena M, Ruipérez-Pacheco E, Gil-Prados I, García-Santos J, Bellón-Del Amo M, Hernández-Aguado JJ, de la Fuente Valero J, Zapardiel I, Coronado Martín PJ. Effectiveness of Prophylactic Human Papillomavirus Vaccine in the Prevention of Recurrence in Women Conized for HSIL/CIN 2-3: The VENUS Study. Vaccines (Basel). 2022 Feb 14;10(2):288. doi: 10.3390/vaccines10020288. PMID: 35214747; PMCID: PMC8879017. http://pubmed.ncbi.nlm.nih.gov/35214747/

 Di Donato V, Caruso G, Bogani G, Cavallari EN, Palaia G, Perniola G, Ralli M, Sorrenti S, Romeo U, Pernazza A, Pierangeli A, Clementi I, Mingoli A, Cassoni A, Tanzi F, Cuccu I, Recine N, Mancino P, de Vincentiis M, Valentini V, d'Ettorre G, Della Rocca C, Mastroianni CM, Antonelli G, Polimeni A, Muzii L, Palaia I. HPV Vaccination after Primary Treatment of HPV-Related Disease across Different Organ Sites: A Multidisciplinary Comprehensive Review and Meta-Analysis. Vaccines (Basel). 2022 Feb 4;10(2):239. doi: 10.3390/vaccines10020239. PMID: 35214697; PMCID: PMC8879645. http://pubmed.ncbi.nlm.nih.gov/35214697

Answer: To comply with the reviewer’s comment we modified the sentence, accordingly.

References were added, accordingly. 

Round 2

Reviewer 1 Report

I have no further comments.